# Sorption Thermodynamics of CO_2_, H_2_O, and CH_3_OH in a Glassy Polyetherimide: A Molecular Perspective

**DOI:** 10.3390/membranes9020023

**Published:** 2019-02-01

**Authors:** Giuseppe Mensitieri, Giuseppe Scherillo, Pietro La Manna, Pellegrino Musto

**Affiliations:** 1Department of Chemical, Materials and PrNoduction Engineering, University of Naples Federico II, Piazzale Tecchio 80, 80125 Naples, Italy; gscheril@unina.it; 2Institute for Polymers, Composites and Biomaterials, National Research Council of Italy, Viale Campi Flegrei 34, 80078 Pozzuoli (Na), Italy; pietro.lamanna@unina.it (P.L.M.); pellegrino.musto@cnr.it (P.M.)

**Keywords:** glassy polymers, sorption thermodynamics, lattice fluid theory, polyetherimide, carbon dioxide, water, methanol

## Abstract

In this paper, the sorption thermodynamics of low-molecular-weight penetrants in a glassy polyetherimide, endowed with specific interactions, is addressed by combining an experimental approach based on vibrational spectroscopy with thermodynamics modeling. This modeling approach is based on the extension of equilibrium theories to the out-of-equilibrium glassy state. Specific interactions are accounted for in the framework of a compressible lattice fluid theory. In particular, the sorption of carbon dioxide, water, and methanol is illustrated, exploiting the wealth of information gathered at a molecular level from Fourier-transform infrared (FTIR) spectroscopy to tailor thermodynamics modeling. The investigated penetrants display a different interacting characteristic with respect to the polymer substrate, which reflects itself in the sorption thermodynamics. For the specific case of water, the outcomes from molecular dynamics simulations are compared with the results of the present analysis.

## 1. Introduction

Sorption thermodynamics of low-molecular-weight compounds within synthetic polymers, as well as the associated interactional issues, is a subject of considerable technological and fundamental interest, as proven by the recent literature reporting on experimental characterization techniques, theoretical approaches, and molecular simulations [1,2,3,4,5,6,7,8,9,10,11,12,13]. Sorption of gases and vapors in polymers has important technological implications as, for example, durability of polymer matrices for composites [14], membranes for separation of liquid and gaseous mixtures [15,16,17,18], and barrier polymers for packaging applications [19].

Depending on the chemical structure of the polymer and the penetrant, the system can be endowed with specific interactions, such as hydrogen bonding or other acid–base type interactions. We previously investigated and theoretically modeled sorption thermodynamics of gases and vapors in glassy polymers characterized by different levels of interactions [20,21,22,23] by complementing the theoretical modeling with the wealth of information gathered from vibrational spectroscopy and gravimetric measurements. These techniques provided a comprehensive molecular characterization of the systems under scrutiny, which was used to tailor the structure of the model. In detail, the experimental results were successfully interpreted using the so-called non-equilibrium theory for glassy polymers with non-random hydrogen bonding (NETGP-NRHB) [22], a model based on a compressible lattice fluid framework which is able to account both for the specific interactions and the non-equilibrium nature of glassy polymers. This approach is based on an extension of the equilibrium NRHB equation of state (EoS) theory [24,25], originally developed by Panayiotou et al. to deal with rubbery polymers. This extension was performed following the procedure introduced by Sarti and Doghieri [26,27].

Polyimides find relevant applications as membrane materials in view of their permeability and selectivity properties, as well as of their strong resistance to organic solvents [28]. Polyetherimide (PEI) belongs to this class of polymers. It is an amorphous engineering thermoplastic with a relatively high glass transition temperature (≅216 °C), a chain flexibility provided by ether groups, and high heat resistance imparted by aromatic imide groups. Thanks to its excellent chemical, mechanical, thermal, and transport properties, and to its durability [29,30,31], it is the material of choice for several separation processes [19,32,33,34,35]. Understanding at a molecular level the interactions between polymer backbone and penetrants is crucial for the optimal design of separation processes. In particular, hydrogen-bonding (HB) interactions significantly affect membrane mass transport and separation performances in those processes involving water and alcohols [36,37,38,39,40,41,42,43,44,45].

We present here a theoretical analysis of the sorption of low-molecular-weight penetrants in PEI combined with experimental investigations. The present contribution puts together unpublished results on the CO_2_–PEI system with results on the H_2_O–PEI and CH_3_OH–PEI systems obtained by our group and already reported in recent publications [20,46,47,48]. The aim was to compare the thermodynamic behavior of systems displaying widely different interactive characteristics. As anticipated, non-equilibrium lattice fluid models are combined with Fourier-transform infrared (FTIR) spectroscopy and gravimetric analyses. Information gathered from the experiments was used to identify the involved interactions to be accounted for in the modeling and to validate model predictions. A rather clear physical picture emerged on self- and cross-interactions, providing quantitative and qualitative indications on the involvement of the moieties present on the polymer backbone.

## 2. Materials and Methods

### 2.1. Materials

Totally amorphous PEI (see Scheme 1) was a commercial-grade product, kindly supplied by Goodfellow Co., PA, USA, in the form of a 50.0-μm-thick film. To obtain film thicknesses suitable for FTIR spectroscopy, the original product was first dissolved in chloroform (15% *w*/*w* concentration) and then cast onto a tempered glass support. The solution was spread using a calibrated Gardner knife, which allows one to control the film thickness in the range of 10–70 μm. The cast film was dried 1 h at room temperature and 1 h at 80 °C to allow most of the solvent to evaporate, and at 120 °C under vacuum over night. At the end of the drying protocol, the film was removed from the glass substrate by immersion in distilled water at 80 °C.

Thinner films (1.0–3.0 μm) were prepared via a two-step, spin-coating process performed with a Chemat KW-4A apparatus from Chemat Technologies Inc (Northridge, CA, USA). Spinning conditions were 12 s at 700 rpm for the first step and 20 s at 1500 rpm for the second step. The spin-coated films were dried in the same conditions as for the thicker films, and freestanding samples were removed in distilled water at room temperature.

Methanol used for sorption experiments was purchased from Sigma-Aldrich (Milano, Italy) with purity higher than 99.6%. Deionized water was used for water vapor sorption experiments. Before use, both water and methanol were degassed through several freezing–thawing cycles. CO_2_ with a purity of 99.5% was purchased from Linde, Germany.

### 2.2. Gravimetric Measurements

The apparatus used to determine the sorption isotherms in PEI was based on the measurement of the elongation of a calibrated quartz spring microbalance (see Figure 1). The microbalance system recorded the weight gain of a sample exposed to a controlled environment (temperature and pressure of gaseous CO_2_ or of water and methanol vapor) The quartz spring was placed in a water-jacketed glass cell that was connected, through service lines, to a CO_2_ cylinder or to a water or methanol reservoir, to a turbo-molecular vacuum pump, and to pressure transducers. The polymer sample hung from the hook of the quartz spring. Changes in weight of the sample determined by sorption of the penetrant brought about the elongation of the spring that was monitored using a traveling camera, equipped with a macro objective, screwed on a computer-controlled piezoelectric slide. The microbalance provided a sensitivity of 1.0 μg with an accuracy of ±2.0 μg.

Gravimetric sorption isotherms were collected at 18, 27, and 35 °C in the case of CO_2_, at 30, 45, 60, and 70 °C in the case of water, and at 30 °C in the case of methanol. Sorption tests were performed starting from neat polymer by increasing the pressure of penetrant in a stepwise manner, waiting for the attainment of a constant weight at each step. Full details about the experimental procedure are given elsewhere [20,21,22].

### 2.3. In Situ FTIR Spectroscopy

Spectroscopic analysis of sorption processes was performed by time-resolved acquisition of the spectra of polymer films exposed to gaseous carbon dioxide or water and methanol vapors, at a constant pressure. These measurements were carried out in the transmission mode, and sorption kinetics was monitored up to the attainment of equilibrium.

The spectroscopic set-up consisted of a diffusion cell placed in the sample compartment of a suitably modified FTIR spectrometer. In the case of CO_2_ sorption, the tests were performed with flowing gas. A schematic view of the apparatus is reported in Figure 2. The cell was connected to a gas line where the flux was regulated by a mass-flow controller and the downstream pressure was controlled by a solenoid valve. The cell temperature could be controlled between −190 °C and 350 °C. Differential sorption tests at 35 °C were performed by increasing stepwise the CO_2_ pressure within the 40–150 Torr range.

In the case of water and methanol vapors, the tests were performed in static conditions. A custom-designed vacuum-tight FTIR cell was used to acquire the time-resolved FTIR spectra during the sorption experiments (see Figure 3). The cell, whose temperature could be controlled between −20 °C and 120 °C, positioned in the sample compartment of the spectrometer, was connected through service lines, to a water or methanol reservoir, a turbo-molecular vacuum pump, and pressure transducers. Full details of the experimental set-up are reported in References [49,50]. Before each sorption measurement, the sample was dried under vacuum overnight at the test temperature in the same apparatus used for the test. Differential sorption tests at 30 °C were performed by increasing stepwise the relative pressure of the vapor, p/p_0_ (where p is the pressure of the vapor and p_0_ is the vapor pressure at the test temperature), within the 0.0–0.6 range.

Spectra were collected using a Spectrum 100 FTIR spectrometer (PerkinElmer, Norwalk, CT, USA), equipped with a Ge/KBr beam splitter and a wide-band deuterated triglycine sulfate (DTGS) detector. The spectral resolution was set to 2 cm^−1^, with an optical path difference (OPD) velocity of 0.5 cm/s. Collection time for a single spectrum was 2.0 s. Spectra were collected and stored for further processing in the single-beam mode. A dedicated software package was employed to control automated data acquisition (*Timebase* from Perkin-Elmer).

The cell without sample, at the test conditions, was used as background to obtain full absorbance spectra (i.e., polymer plus sorbed compound). The spectra representative of the different penetrants were isolated by difference spectroscopy (DC), which allowed us to eliminate the interference of the polymer substrate.
*A_d_*(*v*) = *A_s_*(*v*) − *kA_r_*(*v*),
where *v* is the frequency, and *A_d_(v)*, *A_s_(v)*, and *A_r_(v)* are, respectively, the difference spectrum (the sorbed penetrant), the sample spectrum (the polymer containing the sorbed penetrant), and the reference spectrum (the dry polymer). The subtraction factor, *k*, allows to compensate for changes in the optical path (sample thickness) resulting from possible swelling. In the present cases, it was verified by spectroscopic means that a negligible volume change occurred; hence, a *k* value of 1 was consistently taken.

In the transmission FTIR measurements, sample thickness could be used to adjust the sensitivity, provided that the polymer substrate made no or limited interference in the frequency range where the relevant signals of the penetrant spectrum were located. Obviously, the sensitivity increment was achieved at the expense of the time to equilibration, and a trade-off was established between sensitivity and experiment duration. For this reason, thick samples (40–60 μm) were used to monitor the spectrum of the penetrant, whose concentration was very low, thus achieving an optimum signal-to-noise ratio. Conversely, to investigate the effect of the penetrant on the polymer spectrum, thin films were used (1–3 μm) that allowed keeping the intense carbonyl bands of PEI in the range of linearity of the absorbance vs. concentration curve.

## 3. Theoretical Background: Modeling of Sorption Thermodynamics

Sorption thermodynamics of low-molecular-weight compounds in glassy polymers was addressed using different approaches, some of which account for the molecular details associated with this process. In fact, there are various sorption modes, occurring simultaneously, that should be considered in modeling sorption equilibria: bulk dissolution and specific interactions (including self and cross hydrogen bonding, Lewis acid/Lewis base interactions, and penetrant clustering). Another relevant issue, when dealing with glassy polymers, is their non-equilibrium nature. In this section, we provide an overview of some significant lattice fluid modeling approaches proposed in the literature to interpret sorption thermodynamics in rubbery and glassy polymers. The reader is referred to Appendix A for the mathematical details of some theoretical approaches relevant in the present context.

In the last four decades, EoS-based theoretical models were introduced to address equilibrium thermodynamics of rubbery polymer–penetrant mixtures, possibly accounting for the occurrence of specific interactions (e.g., hydrogen bonding (HB) and Lewis acid/Lewis base interactions). Sorption thermodynamics in glassy polymers, endowed or not with specific interactions, was successfully interpreted by properly extending the equilibrium models, originally developed to describe sorption thermodynamics in rubbery polymers, to the case of polymers in an out-of-equilibrium glassy state.

The compressible lattice fluid (LF) model proposed by Sanchez–Lacombe (SL) [51,52,53] is one of the early examples of EoS-based approaches that were developed to describe sorption in rubbery polymers. Such a model assumes that a random mixing occurs within the polymer–penetrant mixture. The lattice fluid framework for fluid mixtures takes each component (i.e., polymer and low-molecular-weight penetrant in the present context) as consisting of a series of chemically bonded -mers, each one accommodated in a cell of the lattice. Since the lattice is compressible, not all the cells are occupied by the -mers, some of them being possibly empty (holes). Chemical bonds are established between first neighbor -mers belonging to the same molecule, but contacts are also established with other non-bonded -mers located in adjacent first-neighbor cells. The coordination number, *z*, which is a parameter of the model lattice, rules the number of contacts between first-neighbor cells. Contacts established between -mers of the same type are named homogeneous; otherwise, they are said to be heterogeneous. Sorption equilibrium in binary systems is ruled by the equivalence of chemical potentials of each component in the two phases in contact, i.e., the polymer–penetrant mixture and the gaseous phase. The SL theory provides the expressions for the chemical potentials of each of the mixture components, as well as the expression of the equations of state of the mixture and of each pure component. Since it is generally assumed that no polymer is present in the gaseous phase, the equivalence is only imposed to the chemical potentials of the penetrant in the two phases in contact. Densities of the two phases in contact are provided by the EoS of each phase. Parameters of the model are the three scaling parameters of the EoS for the two pure components and the value of the binary interaction parameter that is related to the energy associated to formation of heterogeneous contacts occurring between the -mers of the two components in a binary mixture and the energy associated to the formation of homogeneous contacts between the -mers of each component.

Later, lattice fluid theories were further developed to describe sorption of low-molecular-weight compounds in rubbery polymers for systems endowed with HB interactions. These approaches were developed by properly modifying available LF models that account only for mean field interactions to include the effect of occurrence of self and cross hydrogen bonding. Worthy of mention is the model proposed by Panayiotou and Sanchez (PS) [54] that, starting from the original Sanchez–Lacombe theory, introduced additional terms accounting for the formation of HB interactions. A key assumption in this development was the factorization of the configurational partition function in two separate contributions, respectively associated with mean field interactions and to specific interactions. The PS model, in fact, combines the SL mean field contribution with an HB contribution formulated on the basis of a combinatorial approach first proposed by Veytsman [55,56]. This model requires, in addition to the parameters already mentioned in the case of SL theory, the HB interaction parameters associated with each type of proton donor–proton acceptor adduct that can be formed in the system.

A remarkable intrinsic simplification of the SL model, as well as of the related PS model, is that the evaluation of the mean field contribution is, in both cases, based on a random arrangement of r-mers and holes in the lattice. This assumption is that a more inappropriate arrangement suggests a greater number of non-athermal contacts occurring between different kinds of r-mers [57]. Several approaches were then proposed to account for possible non-random distribution of contacts by following the pioneering work of Guggenheim [58]. As a first step, systems without specific interactions were addressed. Also in this case, the assumption was that the partition function can be factorized, in an ideal random contribution and in a non-random contribution “correction term”. The latter was formulated considering the establishment of each contact as a reversible chemical reaction (the so-called “quasi-chemical approximation”). The original development proposed by Guggenheim was based on a lattice fluid system without holes, but Panayiotou and Vera (PV) improved this theory by introducing a compressible LF model (i.e., with the presence of hole sites) [59]. In the PV model, only contacts between -mers of the mixture components were non-random, while the hole site distribution was still random. Later, You et al. [60] and Taimoori and Panayiotou [61] further upgraded this approach introducing the non-randomness of all the possible couples of contacts, still adopting a non-random quasi-chemical approximation.

A further step consisted in the formulation of an LF-EoS theory accounting both for the occurrence of specific interactions and for the non-randomness of distribution of contacts within the lattice. In fact, Yeom et al. [62] and Panayiotou et al. [2,24,25,63,64] modified the PV approach to include the contribution of HB interactions. Again, the key assumption was the factorization of the configurational partition function in different terms: a mean field random term, a non-randomness “correction term”, and a term accounting for specific interactions. In the present context, the non-random model is of particular interest, which also accounts for cross and self HB interactions, proposed by the group of Panayiotou in References [2,24,25]. We refer to this theory hereafter as the “non-random lattice fluid hydrogen bonding” (NRHB) model. The theory can be used to calculate the chemical potential of the polymer and of the penetrant, both in the polymer mixture and in the vapor or gaseous phase in equilibrium with it, as well as the EoS of both phases. The types of parameters required to perform calculations are similar to those required in the case of PS model. Examples of good correlations of the experimental sorption isotherms using the NRHB model were provided by Tsivintzelis at al. [65] for the case of mixtures of poly(ethylene glycol), poly(propylene glycol), poly(vinyl alcohol), and poly(vinyl acetate) with several solvents, including water.

The theories discussed so far are well suited to describe the sorption behavior of rubbery polymers. When dealing with modeling of sorption thermodynamics in a glassy polymer, one has to tackle an additional complexity, i.e., the non-equilibrium nature of the glassy state. Rational non-equilibrium thermodynamics provides an adequate theoretical framework to address this issue. In fact, using a proper reformulation, it is possible to extend the equilibrium theories developed for mixture thermodynamics—adequate for describing sorption thermodynamics in rubbery polymers—to the case of sorption in glassy polymers. This procedure is based on thermodynamics endowed with internal state variables. Order parameters that mark the departure, at a certain pressure and temperature, from the equilibrium conditions, are selected to play the role of internal state variables. In fact, in addition to the external state variables, which are the only ones needed to describe the state of a system at equilibrium (e.g., pressure, temperature, and concentration), a set of order parameters are used as internal state variables to provide a proper interpretation of the state of non-equilibrium glassy polymer–penetrant mixtures. This line of thought was introduced by Sarti and Doghieri [26,27], which adopted the density of the polymer in the mixture as the only order parameter and internal state variable. These authors introduced a procedure that enables the extension to non-equilibrium glassy systems of several equilibrium statistical thermodynamics mixture theories (in Appendix A, some further details are provided). This approach, referred to as “non-equilibrium thermodynamics for glassy polymers” (NETGP) [66], was demonstrated to be very effective in describing the thermodynamics of numerous binary and ternary glassy polymer–penetrant mixtures. A relevant difficulty that one encounters when applying this theory is that the expression for evolution kinetics of the internal state variable has to be known. To avoid dealing with complex evolution kinetics of non-equilibrium polymer density, a “simplified” version of this approach considers the polymer as being “frozen” in a kinetically locked pseudo-equilibrium (PE) state. This assumption is legitimate for glassy polymers at temperatures well below their glass transition temperature and when the concentration of penetrant within the polymer phase is low. To deal with cases in which such conditions do not occur, the theory was properly formulated to take also into account the structural evolution of the glassy system during penetrant sorption. The NETGP approach requires the knowledge of the same parameters as for the equilibrium theory from which it is derived. The phase equilibrium is still dictated by the equivalence of chemical potentials of each component in the two phases. However, differently from the case of sorption in rubbery polymers, no equation of state can be written for the polymer phase. In fact, in the case of glassy polymers, the value of polymer density cannot be calculated using an equilibrium EoS, but is determined by its intrinsic evolution kinetics; its value, which is necessary to perform model calculations, should be known from independent experimental data. In particular, in the present context, we refer to the so called “non-equilibrium lattice fluid” (NELF) model [26,27] as the NETGP extension of the equilibrium Sanchez–Lacombe theory to treat sorption in glassy polymers not endowed with specific interactions (see Appendix A, for details).

To deal with sorption thermodynamics for glassy polymer–penetrant systems exhibiting specific interactions, extensions of equilibrium models able to cope with self- and cross-interactions were then proposed. Referring to the case of EoS-based models, of particular relevance are the extensions of “statistical associating fluid theory” (SAFT) [67] and NRHB [68] approaches. Our group developed and applied an extension of the NRHB model to non-equilibrium glassy polymers to account for HB interactions in water/glassy polymer mixtures [22]. We refer to this model here as NETGP-NRHB (see Appendix A for details). Again, this model requires the same parameters as for the equilibrium NRHB model. In addition, the value of polymer density should also be known, since it cannot be provided by an EoS.

## 4. Results and Discussion

In this section, we discuss the results of experimental analysis and of theoretical modeling of the PEI–CO_2_, PEI–water, and PEI–methanol systems. Results of in situ FTIR spectroscopy are presented along with their elaboration by means of two-dimensional correlation spectroscopy (2D-COS) [23,69,70,71]. Indications emerging from FTIR measurements were used to tailor the structure of sorption thermodynamics model, which was then used to fit gravimetric sorption isotherms.

### 4.1. PEI–CO_2_ System

#### 4.1.1. FTIR Analysis: Absorbance, Difference, and Two-Dimensional Correlation Spectra

As reported in Section 2.3, to achieve an optimum signal-to-noise ratio, thick samples were used to monitor the spectrum of the penetrant, whose concentration was very low. Conversely, to analyze the effect of the sorbed penetrant on the polymer spectrum, thin films were used in order to keep the relevant bands of PEI in the range of linearity of the absorbance vs. concentration curve. In Figure 4A, the spectra of a fully dried film (red trace) and of the same film equilibrated at 35 °C under a CO_2_ pressure of 150 Torr (blue trace), collected on a 37.7-μm-thick film, are compared. This was the kind of sample used to perform the subtraction spectroscopy analysis on the spectral region populated by signals associated to penetrant molecules. In Figure 4B, the same comparison for the spectra collected on a much thinner film, with a thickness of 2.6 μm, is shown. This was the kind of sample used to perform the subtraction spectroscopy analysis in the spectral region with signals characteristic of the polymer, in order to detect possible perturbations related to the presence of sorbed CO_2_. In particular, attention was focused on the intense carbonyl bands of PEI, and the thickness was selected to keep the associated absorbance vs. concentration curve in the range of linearity. It is worth noting that, in the case of the “thin” sample, the weakness of the penetrant signals, even at the maximum CO_2_ pressure, did not allow a reliable band-shape analysis. Furthermore, the sorption kinetics was so fast that a time-resolved measurement and the associated two-dimensional correlation analysis (vide infra) were unfeasible.

It was immediately apparent, from the comparison of the two traces reported in Figure 4A, that sorbed carbon dioxide produced two signals, centered at 2336 and 655 cm^−1^. The high-frequency band was due to the antisymmetric stretching mode (ν_3_) at 2336 cm^−1^ plus a satellite peak at 2324 cm^−1^. The latter component was a non-fundamental transition, in particular a hot band enhanced by Fermi resonance with the main stretching mode [72]. It is to be stressed that the peak at 2324 cm^−1^ was not due to a second CO_2_ species, but to a further signal produced by the same species that generated the main absorption at 2336 cm^−1^.

The ν_3_ mode was well suited for quantitative analysis and for achieving information at the molecular level. The carbon dioxide bending mode (ν_2_) found at around 655 cm^−1^ was less useful, being much weaker and superimposed onto intense polymer bands. The suppression of the polymer matrix interference by difference spectroscopy [50,69] allows one to isolate the spectrum of absorbed CO_2_. The integrated absorbance of the 2336 cm^−1^ band was collected at each pressure as a function of sorption time up to attainment of sorption equilibrium.

In Figure 5A, the ν_3_ band of CO_2_ after sorption equilibrium at different pressures is reported. The correlation between integrated absorbance and gravimetric data, collected at 35 °C, at sorption equilibrium at different CO_2_ pressures, is represented in Figure 5B. The behavior was linear through the origin, thus verifying the Lambert–Beer law, which allows one to convert the absorbance data into absolute concentration values. In the explored pressure range (40–160 Torr), the sorption isotherm was linear (see Figure 5C). The band-shape of the ν_3_ mode was exactly coincident at all pressures (see Figure 5A) indicating that, in the explored range, the molecular interactions formed with the polymer substrate did not depend on CO_2_ concentration.

The time-resolved spectra were subjected to 2D-COS analysis, which is very effective for investigating molecular interactions [69]. This is a perturbative technique applied to systems that are initially at equilibrium; these are then subjected to an external stimulus and the spectral response of the system is treated by a correlation analysis. The covariance of two correlated signals (peak absorbance, in the present case) is measured as a function of a third common variable related to the perturbing function (time, in the present case). This procedure spreads the spectral data over a second frequency axis and allows unambiguous assignments through the correlation of bands. It produces synchronous and asynchronous maps. In the latter, the correlation intensity for signals evolving at the same rate vanishes [71], thus providing a resolution enhancement. Moreover, valuable dynamic information can be also gathered.

In Figure 6A,B, the synchronous spectrum in the 2250–2420 cm^−1^ range, obtained from the time-resolved spectra collected during the sorption experiment at 150 Torr and 308 K, and the power spectrum, i.e., the autocorrelation profile taken across the main diagonal, are reported. The synchronous spectrum (Figure 6A) displays a highly characteristic cross-shape; in the power spectrum (Figure 6B), a strong auto-peak was detected at 2336 cm^−1^, corresponding to the main component in the frequency spectrum, plus a weak, fully resolved feature at 2324 cm^−1^. The off-diagonal wings present in the synchronous map (Figure 6A) extend by about 60 cm^−1^ and reflect the presence of a further component, increasing on sorption or decreasing on desorption concurrently with the main peaks (at 2336 and 2324 cm^−1^). The elongated shape of the off-diagonal feature indicated that this component was significantly broader than the main peaks.

The asynchronous spectrum was featureless (see Figure 7A,B). Only noise was present despite the significant intensity of the band and its complex structure. This result was relevant; it demonstrated that all the components in the ν_3_ profile evolved synchronously, which, in turn, signified that the species they originated from had comparable dynamics or, alternatively, that a single molecular species produced all the observed components. Albeit no conclusive evidence is yet available, we are inclined toward the second hypothesis, in view of previous studies on solvated low-molecular-weight compounds [73,74].

Similar band profiles were reported in the literature (i.e., a broad Gaussian component superimposed onto a sharper peak of larger intensity) and were interpreted assuming the Gaussian part as being a remnant of the gas-phase spectrum, while the sharp, Lorentz-like component was interpreted as the purely vibrational transition active in the condensed phase (but not in the gas-phase) [73,74]. According to this interpretation, the two-component band-shape was the consequence of probe dynamics within the molecular environment (essentially free rotation in the early stages of the relaxation process, 0.2–1.0 ps), which produced the Gaussian part, and a random rotational diffusion regime at later stages (the so-called Debye regime), giving rise to the Lorentzian part. The band-shape models based on small-molecule mobility invariably tend to overlook the role played by molecular interactions. This is partially due to the difficulty in embracing the effects of specific contacts in a random collisional framework. In the present case, the specific interaction between CO_2_ and PEI emerged clearly from the vibrational analysis (vide infra) and was taken into account when interpreting the band-shape and the 2D-COS results. Our view is that the probe molecules sorbed in PEI were characterized by very short free-rotation regimes (the Gaussian component is barely detectable in the frequency spectrum); the main component was due to a single molecular species of CO_2_ forming a specific type of interaction with the polymer substrate. The uniqueness of the interaction is suggested by the sharp and highly symmetrical nature of the main ν_3_ component, coupled with the absence of asynchronous features in the 2D-COS map. Work is in progress to substantiate this hypothesis and to put the above considerations on more quantitative grounds.

To complete the characterization of the probe-to-substrate interaction, we had to recognize the functional group(s) of the PEI backbone that was (were) involved in the formation of the adduct with the penetrant. In fact, there were several possible candidates, such as the imide carbonyls, the ether oxygens and, to a lesser extent, the aromatic rings. The analysis was performed by detecting the perturbation brought about by the probe to the spectrum of the substrate in terms of peak shifts and/or band-shape distortion, effects that were subsequently interpreted in terms of geometry and electron-density distribution of the molecular aggregate. The reversibility of the observed effects was also assessed, i.e., the obtainment of the unperturbed spectrum when the probe was fully desorbed. Measurements of this kind require films in the thickness range of 1.0–3.0 μm, in order to maintain the whole spectrum within the range of absorbance linearity. In the present case, these films were prepared ad hoc via a spin-coating process. Since the interaction was weak, the effects were expected to be very subtle; in these circumstances, difference spectroscopy was demonstrated to be a viable approach [46]. In Figure 8, the difference spectra in the frequency range 1820–1660 cm^−1^ are reported, obtained by subtracting the spectrum of the fully dried sample from the spectra of the polymer sample equilibrated with gaseous CO_2_ at a pressure of 150 Torr and at the different temperatures. It is explicitly noted that the reference and the sample spectra to be subtracted were collected at the same temperature to avoid any spurious perturbation of the vibrational response. Lowering the temperature is expected to narrow the peaks while increasing the separation among the components, which should enhance the shift effect. An increase in the number of interacting CO_2_ molecules is also possible at lower temperatures, as a consequence of the reduced kinetic energy of the probe.

The difference spectra in Figure 8 display the typical first-derivative features associated with red-shifts, that is, a negative lobe preceding the positive. Thus, a lowering of the peak frequency was produced in the CO_2_-containing samples. The effect was evident for both peaks occurring in the 1800–1680 cm^−1^ interval, according to the nature of the corresponding vibrational modes, i.e., the in-phase, ν_ip_(C=O), and out-of-phase stretching, ν_oop_(C=O), of the imide carbonyls. The two-lobe profiles were symmetric, i.e., the positive and the negative components had comparable intensities and grew progressively as the temperature decreased. Furthermore, the effect was fully reversible, as demonstrated by comparing the reference spectrum with that of the sample equilibrated at *p* = 150 Torr and subsequently desorbed. These results confirm that the observed shifts, albeit very subtle (maximum shift of the main carbonyl component = 0.25 cm^−1^), actually originated from the probe/substrate interactions and that these interactions were weak. For comparison, we recall that the red-shift caused by the interaction of the same carbonyl groups with water molecules (H-bonding) amounted to 0.8 cm^−1^ [46]. All difference spectra were featureless below 1250 cm^−1^. The absorbance spectrum of PEI displays two intense bands at 1238 and 1215 cm^−1^, involving significant contributions from the C–O–C stretching modes [46]. The fact that these bands (as well as aromatic peaks) remained unaffected indicates that both the ether oxygens and the aromatic rings did not “see” the presence of the CO_2_ molecules. The effects discussed above clarify the interaction mechanism: the imide carbonyls were selectively involved in a way that produced a lowering of the C=O force constant. The carbon atom of the carbon dioxide molecule, which brought a partial positive charge as a consequence of the two oxygens to which it was bound, formed a weak Lewis acid/Lewis base interaction with the imide carbonyls, as schematically represented in Figure 9. The above conclusions are in full agreement with the 2D-COS analysis.

#### 4.1.2. Modeling Sorption Thermodynamics

The FTIR analysis presented in the previous section indicates that carbon dioxide molecules absorbed within PEI tended to establish weak Lewis acid/Lewis base interactions with carbonyls along the polymer backbone. The interactional energy was close to a mean field value; hence, for the sake of a macroscopic thermodynamics description of the system, it was pointless to introduce distinct specific interaction terms in the model (as for the NETGP-NRHB model). Based on these premises, the NELF approach (see Appendix A, for relevant equations) was used, which provided a good interpretation of gravimetric sorption isotherms. In Figure 10, the experimental gravimetric sorption isotherms, along with the curves obtained by a concurrent fitting of the data with the NELF model at three temperatures (18, 27, and 35 °C), are reported. Only one fitting parameter, i.e., the Sanchez–Lacombe binary interaction parameter, *χ*_12_ (see Appendix A, for its definition), was used. A value of *χ*_12_ = 0.0313 ± 0.003 was estimated, thus pointing to a limited deviation from the geometric mixing rule for characteristic pressure (see Equation (A24)). The scaling parameters for the SL EoS of pure PEI were calculated by fitting pressure, volume, and temperature (PVT) data available from a previous investigation [20] with the SL-EoS model, while those of pure CO_2_ were taken from Reference [26]. The (non-equilibrium) density values of neat PEI (i.e., *ρ*_2,∞_ as defined in Appendix A) at each temperature (assumed here to be time-independent) used in the model were taken from the literature [20]. All parameters are reported in Table 1 and Table 2.

### 4.2. PEI–H_2_O System

#### 4.2.1. FTIR Analysis: Absorbance, Difference, and Two-Dimensional Correlation Spectra

For the analysis of water absorbed in PEI, we considered the normal modes of the water molecule in the ν(OH) frequency range (3800–3200 cm^−1^). Other regions of potential interest for the spectroscopic analysis were not available for the case at hand due the exceedingly low intensity of the signals. This system was discussed in a recent publication [46], and we present here the most relevant results.

In Figure 11A, the difference spectra in the OH stretching region (ν(OH)) are reported for the sample equilibrated with water vapor at different p/p_0_ values. The band was representative of sorbed water and exhibited a profile with two maxima at 3655 and 3562 cm^−1^. The well-resolved multicomponent band-shape clearly indicated the occurrences of different water species in contrast to the situation observed in the case of CO_2_, where a single component pattern was detected.

To improve the resolution and to deepen the spectral interpretation, 2D-COS was also carried out in this case; the time-resolved spectra collected during a water sorption experiment performed at a p/p_0_ = 0.6 were used for this purpose. The resulting asynchronous spectrum is reported in Figure 11B. The rich pattern contrasts with the featureless map detected for the CO_2_–PEI system (compare Figure 11B with Figure 7A), thus confirming that asynchronous 2D-COS spectra are a powerful and sensitive tool to highlight complex molecular interaction scenarios. In particular, in the case of the H_2_O–PEI system, it emerged that there existed two couples of signals, suggesting the presence of two distinct water species. In detail, the two sharp peaks at 3655–3562 cm^−1^ were assignedto the asymmetric and the symmetric stretching modes (ν_as_ and ν_s_), respectively, of isolated water molecules interacting via H-bonding with the PEI backbone (cross-associated or first-shell water molecules). The second doublet at 3611–3486 cm^−1^ was related to water molecules self-interacting with the first shell species (self-associated or second-shell water molecules).

To identify the active sites of polymer backbone involved in the specific interaction with first-shell water molecules, an analysis was performed on thin samples (thickness in the range 1.0–3.0 μm). The red-shift of the ν_ip_(C=O) and ν_oop_(C=O) modes demonstrated the involvement of the imide carbonyls as proton acceptors in H-bonding. The peaks of the aryl ether groups remained unperturbed, which demonstrated that their interaction with water was negligible, if present. In addition, it was found that the largely prevalent stoichiometry of the first-shell adduct was 2:1, i.e., a single water molecule bridged two carbonyls: –C=O---H–O–H---O=C–. The two water species identified spectroscopically are schematically represented in Figure 12, where the peak frequencies they produced are also indicated.

Molecular dynamics (MD) calculations performed on this system [46] indicated that, at least in the low-to-intermediate relative pressure range of water vapor, these bridges formed by first-shell water were intramolecular, that is, formed by interacting with two contiguous carbonyl groups present on the same macromolecule. In Figure 13, a snapshot of an MD calculation is reported, showing the typical conformation of first-shell water molecules.

Performing a quantitative analysis of the difference spectra of water, based on the information available from gravimetric measurements at different relative humidity, it was possible to obtain a quantitative estimation of the amount of the two water species identified spectroscopically. Details on the calculation procedure are reported in a previous publication [46].

#### 4.2.2. Modeling Sorption Thermodynamics

In view of the significant HB interactions characterizing the PEI–H_2_O system, the interpretation of sorption thermodynamics of water in glassy PEI was approached using the NETGP-NRHB model (see Appendix A for the relevant equations). Information gathered from the vibrational spectroscopy investigation, illustrated in Section 4.2.1, was exploited to tailor the H-bonding interaction contributions in the NETGP-NRHB model. In fact, only one proton acceptor was assumed to be present on the polymer backbone (i.e., the carbonyl of the imide group) and water molecules were assumed to be involved both in cross and self HB interactions. Moreover, the presence of two proton donors and two proton acceptors was assumed for each water molecule.

Also in this case, as for the PEI–CO_2_ system, the non-equilibrium density of the polymer, at the different temperatures, was assumed to be time-independent and its value was taken to be equal to that of pure PEI. These density values, the values of the NRHB EoS scaling parameters for pure PEI and pure water, and the values of the self HB parameters for pure water were retrieved from the literature [20,75] and are reported in Table 1 and Table 3. Experimental gravimetric water sorption isotherms in PEI were available at four temperatures (30, 45, 60, and 70 °C) [47]. For the meanings of scaling and HB interaction parameters appearing in Table 3, the reader is referred to Appendix A.

Concurrent fitting of the four experimental gravimetric isotherms was performed using the NETGP-NRHB model, and the results are shown in Figure 14. The internal energy of formation of water–polymer cross HB, $E_{12}^{0wp}$, the entropy of formation of water–polymer cross HB, $S_{12}^{0wp}$, and the mean field binary interaction parameter, *K*_12_, were used as fitting parameters (their meanings are discussed in Appendix A). In accordance with the previous relevant literature [65], the volume of formation of water–polymer cross HB, $V_{12}^{0wp}$, was taken to be equal to zero. In Table 4, the values of the model parameters as determined by a best fitting procedure along with their 95% confidence intervals are reported, determined using the Jacobian method implemented in the *nlinfit* and *nlparci* routines of the Matlab^®^ software. As evident in Figure 14, the model provided a very good fitting of the experimental sorption isotherms. Calculations performed to carry out isotherm data fitting using the NETGP-NRHB model also provided a prediction for the amount of self HB (i.e., 1–1, involving only water molecules) and of cross HB (i.e., 1–2, between water molecules and carbonyl groups of PEI) interactions established in the PEI–water mixture equilibrated with water vapor at the different p/p_0_ investigated. In particular, calculations provided the number of moles of HB self-interactions (1–1) and of moles of HB cross-interactions (1–2) per gram of amorphous PEI, indicated as $n_{11}/m_{2}$ and $n_{12}/m_{2}$, respectively. These values predicted using the NETGP-NRHB model, at a temperature of 30 °C, are compared in Figure 15 with the results obtained from FTIR spectroscopy and with the outcomes of MD simulations [46]. The agreement is satisfactory in the whole range of water concentrations.

### 4.3. PEI–CH_3_OH System

#### FTIR Analysis: Absorbance, Difference, and Two-Dimensional Correlation Spectra

For the methanol–PEI system at 30 °C, the ν(OH) band-shape (3650–3200 cm^−1^ range) was simpler than for water–PEI (compare Figure 16A and Figure 11A) [47,48]. A single maximum was observed at 3575 cm^−1^, superimposed to a broader component at lower frequencies. The occurrence of two signals only was confirmed by the asynchronous map (Figure 16B) displaying a single cross-correlation centered at 3441–3575 cm^−1^. The other features appearing in the map were relative to the correlations with the ν(CH) signals and were not relevant for the molecular interaction analysis. Perturbation analysis of the PEI spectrum indicated that the carbonyl groups of the polyimide acted as proton acceptors in the H-bonding interaction with methanol, while the ν(C–O–C) band was unperturbed in the presence of methanol, thus indicating that the involvement of ether linkages in H-bonding with the penetrant could be ruled out.

In the light of all the experimental evidence, the peak at 3575 cm^−1^ was assigned to the OH group of methanol directly to the imide carbonyl, which represents the first-shell layer of the sorbed penetrant. The peak at 3441 cm^−1^ was due to self-associated methanol molecules (dimers), and in particular to the OH group bonded to oxygen of the first-shell species. A schematic representation the H-bonding aggregates established in the CH_3_OH–PEI system is depicted in Figure 17.

As for the case of water, it was possible to quantify the amount of each of the two methanol species, as a function of the relative pressure of the methanol vapor phase at sorption equilibrium, by combining the gravimetric sorption isotherm, reported in Figure 18, with the results available from vibrational spectroscopy (see Figure 19) [47,48].

It was noted that, in the investigated range of relative pressure, the concentration of first-shell methanol, *C_fs_*, was significantly higher than that of second-shell methanol, *C_ss_*, thus indicating that, in this range, the monomer (i.e., a single molecule of methanol linked via HB to a carbonyl group), whose concentration was equal to *C_fs_* − *C_ss_*, was the largely prevailing species while self-associated aggregates were in a smaller amount and were likely to be dimers.

Fitting of the sorption isotherm was attempted both with the NELF and the NETGP-NRHB models assuming a time-independent value of polymer density, but the results were not satisfactory, likely due to a non-negligible structural evolution of PEI during methanol sorption. Since information on the actual value of polymer density was not available, we were not able to properly interpret the results with the illustrated class of lattice fluid models.

## 5. Conclusions

A molecular insight into sorption thermodynamics of low-molecular-weight penetrants in a glassy PEI was gained in the case of three penetrants with different interactive characteristics. To this aim, gravimetric analysis and FTIR spectroscopy were combined with macroscopic thermodynamics modeling.

For the case of sorption of carbon dioxide and water, non-equilibrium compressible lattice fluid theories were successfully used to interpret the experimental results. For the analysis of sorption data of carbon dioxide, an extension to non-equilibrium glassy polymers of the equilibrium Sanchez–Lacombe theory was adopted—the so-called NELF model. Conversely, for the case of water sorption, data were interpreted using an extension to non-equilibrium glassy polymers of the original NRHB approach—the so-called NETGP-NRHB model—to cope with specific H-bonding interactions occurring in the PEI–H_2_O system.

In both cases, information gathered from vibrational spectroscopy was useful to choose the proper model and to tailor its structure. In fact, in the case of carbon dioxide, in view of the weak interactions established with groups present on the polymer backbone, the NELF approach, which does not explicitly account for specific interactions, was well suited to describe sorption thermodynamics. On the other hand, in the case of water, FTIR spectroscopy revealed the occurrence of two water species, one self-interacting via H-bonding and the other cross-interacting with carbonyl groups of PEI. This identification of the interactions to be accounted for in the sorption thermodynamics interpretation allowed us to tailor the structure of the H-bonding terms in the NETGP-NRHB model. In this case, FTIR analysis also provided quantitative information on self and cross HB interactions established within the system, which was in remarkably close agreement with predictions of the adopted theoretical approach.

In the case of methanol, we were not able to identify a suitable theoretical approach, likely due to the effect of the penetrant on the density of the polymer substrate in the time frame of the experiments. In this case, we limited ourselves to the spectroscopic identification of the molecular aggregates and the population analysis.

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
