# Peer review of "Sorption Thermodynamics of CO_2_, H_2_O, and CH_3_OH in a Glassy Polyetherimide: A Molecular Perspective"

_membranes, 2019, doi:10.3390/membranes9020023_

Round 1

Reviewer 1 Report

The authors present a sort of review summary of their previous works regarding the characterization of polyetherimide films as to the sorption of CO2, water and methanol molecules, based in vibrational spectroscopy techniques, and comparing them to the non-equilibrium theory for glassy polymers developed by the group of the university of Bologna. The topic is interesting in the quest of gaining understanding on the interaction of common molecules in polyimide membrane fabrication and gas separations, but the novelty becomes blurred because of the intricate language used by the authors and also several mistakes throughout the manuscript that make reading cryptic. I am sorry if I have not understood well.

Besides it is not clear throughout the text when the authors are mentioning general topics or previous results from their own or other authors because references are not cited next to the sentence but a couple of sentences later or so, or mentioned in a very general manner, as in line 41, line 245, “both interactions”, “specific interactions”, what interactions?, line 319, “previous studies”, what previous studies?, line 430, “results of FTIR analysis reported above”, which one?, line 558, “methanol sorption in PEI has been investigated at 30ºC”, where, here or in literature? Say so and explain why. And so on.

About the thickness of the samples, there is some confusion, because two very different ranges are given, from 1-3 microns to 20-40 microns, which is huge for FTIR measurements. In fact, it seems that the IR spectra in Figure4 are obtained for the thick ones while the rest of the spectra in the manuscript are taken from the 1-3 microns. Can the authors provide experiments with the same thickness so that no saturation occurs in any of the techniques employed and results can provide significant information without experimental errors? In line 136, the author say that the changes in sample thickness during measurement “has been verified in the present case”, without explaining how. Have the authors measured the thickness before and after the experiment, how?

Would not be easier to measure the IR spectra of the thin films and see if the saturation in Figure 4 is not repeated, before measuring 2D-COS spectroscopy and analyzing difference spectra of thin films that cannot be compared with anything in the present work?

In the drawings of the experimental setups in Figures 1-3, where is the membrane in all the cases? It is not clear.

Line 158, “survey?

Line 202, “mers”?

Line 205, and the following, what do the authors mean by “contacts”?

The model terms are repeated several times throughout the text but then hardly discussed in the results and discussion. In line 257, for instance, NETGP-NRHB has been already referred to in the manuscript.

Lines 414-415, the authors first say that “all difference spectra were featureless below 1250 cm-1” and then that “PEI displays two intense bands” in that region, which have not been observed in the present work, but in a previous one. The sentences contradiction make difficult to understand the purpose of this citation and statement. Please revise.

Author Response

Reviewer #1:

The authors present a sort of review summary of their previous works regarding the characterization of polyetherimide films as to the sorption of CO2, water and methanol molecules, based in vibrational spectroscopy techniques, and comparing them to the non-equilibrium theory for glassy polymers developed by the group of the University of Bologna. The topic is interesting in the quest of gaining an understanding on the interaction of common molecules in polyimide membrane fabrication and gas separations, but the novelty becomes blurred because of the intricate language used by the authors and also several mistakes throughout the manuscript that makes reading cryptic. I am sorry if I have not understood well.

Authors reply: 

the manuscript has been revised and rearranged, hopefully in a form that makes all the arguments more understandable. Moreover, at the end of the introduction section the aim of the present contribution is better defined, evidencing that new results related to CO2-PEI systems are compared to previous analyses performed by our group on the H2O –PEI and CH3OH-PEI systems. The revised manuscript now reads: ‘The present contribution puts together unpublished results on the CO2-PEI system with results on the H2O-PEI and CH3OH-PEI systems obtained by our groups and already reported in recent publications [46-49]. The aim is to compare the thermodynamic behavior of systems displaying widely different interactive character with respect to the polymer backbone.’ 

 In addition, a detailed description of adopted models is now reported in an Appendix section in the revised version of the manuscript.

Reviewer #1:

Besides it is not clear throughout the text when the authors are mentioning general topics or previous results from their own or other authors because references are not cited next to the sentence but a couple of sentences later or so, or mentioned in a very general manner, as in line 41, line 245, “both interactions”, “specific interactions”, what interactions?, line 319, “previous studies”, what previous studies?, line 430, “results of FTIR analysis reported above”, which one?, line 558, “methanol sorption in PEI has been investigated at 30ºC”, where, here or in literature? Say so and explain why. And so on.

Authors reply: 

In revising the text we took into account the points raised by the reviewer, adding, where it was the case, proper reference to previous works. In the following our reply to the other points:

a) for the points related to lines 41  the sentence has been rephrased as follows: ‘…based on a compressible lattice fluid that is able to account both for the specific interactions  and the non-equilibrium...' .

b) regarding line 245, this part of the manuscript has been completely rearranged.

c) regarding line 319 references have been made to relevant studies (refs. 76 and 77).

d) regarding the line 430 the sentence has been rephrased as follows: ‘The FTIR analysis discussed in the previous section…’

c) regarding line 558 the sentence has been rephrased as follows ‘A FTIR analysis of CH3OH-PEI system, analogous to those performed on CO2-PEI and H2O-PEI systems, has been previously reported by our group [48, 49]. We summarize here the most relevant results, for the sake of comparison to the other two systems.  In particular, methanol sorption in PEI at 30°C is discussed.’

Reviewer #1:

About the thickness of the samples, there is some confusion, because two very different ranges are given, from 1-3 microns to 20-40 microns, which is huge for FTIR measurements. In fact, it seems that the IR spectra in Figure4 are obtained for the thick ones while the rest of the spectra in the manuscript are taken from the 1-3 microns. Can the authors provide experiments with the same thickness so that no saturation occurs in any of the techniques employed and results can provide significant information without experimental errors? In line 136, the author say that the changes in sample thickness during measurement “has been verified in the present case”, without explaining how. Have the authors measured the thickness before and after the experiment, how?

Authors reply: 

In order to clarify the point raised by the reviewer, in section 2.3 the following paragraph has been added, detailing the reasons for using samples with a different thickness in performing the FTIR analysis: ‘In transmission FTIR measurements, sample thickness can be used to adjust the sensitivity provided that the polymer substrate makes no or limited interference in the analytical frequency range. Obviously, the sensitivity increment is achieved at the expenses of the time to equilibration and a trade-off is to be established between sensitivity and experiment duration. For this reason, thick samples (40-60 μm) have been used to monitor the spectrum of the penetrant, whose concentration is very low, thus achieving an optimum signal to noise ratio. Conversely, to investigate the effect of the penetrant on the polymer spectrum, thin films were used (1-3 μm) that allow to keep the intense carbonyl bands of PEI in the range absorbance vs. concentration linearity.’

It is to be stressed that in the thin films the sorption kinetics is instantaneous and cannot be monitored and used for 2D-COS and, at the same time, the penetrant signals are barely detectable. As a consequence, unfortunately a unique thickness for all the measurements cannot be used. 

Regarding the thickness of the sample, we have verified that the intensity of several polymer peaks (in particular the peaks at 1015, 922 and 627 cm-1) is invariant after reaching sorption equilibrium, in each of the tests performed. In view of the accuracy of spectroscopic measurements, this implies a thickness invariance within at least 1% of the initial value, that for our analysis has been considered negligible.

Reviewer #1:

Would not be easier to measure the IR spectra of the thin films and see if the saturation in Figure 4 is not repeated, before measuring 2D-COS spectroscopy and analyzing difference spectra of thin films that cannot be compared with anything in the present work?

Author reply: 

See answer to the previous point.

Reviewer #1:

In the drawings of the experimental setups in Figures 1-3, where is the membrane in all the cases? It is not clear.

Author reply:

in the captions of figures 1-3 is now indicated where is located the membrane.

Reviewer #1:

Line 158, “survey?

Authors reply:

the text has been amended. Now it reads: ‘In the following, we briefly summarize relevant lattice fluid modeling approaches based on Equation of State theories to interpret sorption thermodynamics in rubbery and glassy polymers.’

Reviewer #1:

Line 202, “mers”? Line 205, and the following, what do the authors mean by “contacts”?

Authors reply:

the meaning of the term ‘mers’ and ‘contacts’, that are commonly adopted in the formulation of lattice fluid models, have been made explicit in the revised version of the manuscript. In particular, the following sentences have been added: ‘It is worth reminding here that in lattice fluid models for mixtures each component (i.e. polymer and low m.w. penetrant in the present context) is assumed to consist of a series of chemically bonded mers, each accommodated in a cell of the lattice. Not all the cells in a lattice are occupied by the mers of the components, some of them being possibly empty (holes). Each mer in a cell establishes chemical bonds with first neighbors mers belonging to the same molecule but also establishes contacts with other non-bonded mers located in adjacent first neighbors cells. The number of contacts depends upon the coordination number of the model lattice. These contacts are said to be homogeneous if established by mers of the same type and are said to be heterogeneous otherwise.’

Reviewer #1:

The model terms are repeated several times throughout the text but then hardly discussed in the results and discussion. In line 257, for instance, NETGP-NRHB has been already referred to in the manuscript.

Authors reply:

the models are now well described in the Appendix section making more clear the meaning of parameters and of different variables.

Reviewer #1:

Lines 414-415, the authors first say that “all difference spectra were featureless below 1250 cm-1” and then that “PEI displays two intense bands” in that region, which have not been observed in the present work, but in a previous one. The sentences contradiction make difficult to understand the purpose of this citation and statement. Please revise.

Authors reply: 

actually, there is no contradiction in the sentence. The absorbance spectrum of PEI displays two intense bands at 1238 and 1215 cm-1. Since these bands are not affected by penetrant sorption, the difference spectrum in this region is featureless and reduces to the base line.

Reviewer 2 Report

This work reports the sorption thermodynamics of low molecular weight penetrants in PEI glassy polymers. The interactions between the absorbent and polymer matrix are addressed by combining an experimental approach with thermodynamics modelling. While the results may provide useful information and understanding for membrane studies in the field of membrane gas separation, the quality of the manuscript is quite poor. First, most of the figures are in very poor quality and Figure 13 even uses different decimal. Second, there are no figures which correspond to FTIR and gravimetric adsorption results of both water and methanol. Besides that, Figure 13 shows that the experiment is uncontrollable. Moreover, the comparison between experimental results and modelling in Figure 14 suggests that the model does not well correspond to the experimental results which opposed the authors’ claim in Figure 13. Lastly, the authors did not provide any mathematical equations which are of utmost importance for this manuscript. Although the reports have been cited in this manuscript, without any explicit indication, it is practically impossible to understand what the authors referred to. Therefore, I would recommend to reject this manuscript.

Author Response

Reviewer #2:

This work reports the sorption thermodynamics of low molecular weight penetrants in PEI glassy polymers. The interactions between the absorbent and polymer matrix are addressed by combining an experimental approach with thermodynamics modeling. While the results may provide useful information and understanding for membrane studies in the field of membrane gas separation, the quality of the manuscript is quite poor.

Authors reply: 

The quality of the manuscript has been improved in terms of language and clarity of sentences (see also answers to reviewer #1). Moreover, an Appendix section has been added to better illustrate the mathematical details and the meaning of the relevant parameters of the models adopted to interpret data.

Reviewer #2

First, most of the figures are in very poor quality and Figure 13 even uses different decimal.

Authors reply: 

The quality of the figures has been improved and the symbol used for decimals in figure 13 has been corrected.

Reviewer #2:

Second, there are no figures which correspond to FTIR and gravimetric adsorption results of both water and methanol.

Authors reply: 

The section related to CO2-PEI system contains experimental results in a greater detail as compared to the other two systems since it is reported for the first time. Less experimental  have been provided for H2O-PEI and CH3OH-PEI systems since these have been reported elsewhere.

Reviewer #2 

Besides that, Figure 13 shows that the experiment is uncontrollable.

Authors reply:

Frankly speaking, we hardly understand this comment. 

Reviewer #2:

Moreover, the comparison between experimental results and modeling in Figure 14 suggests that the model does not well correspond to the experimental results which opposed the authors’ claim in Figure 13.

Authors reply:

the sorption thermodynamics model used to interpret the results, actually provides a very good concurrent fitting of gravimetric data at different temperatures (i.e. amount of water sorbed as a function of pressure at different temperatures) by using only three parameters. Usually, this experimental information is the only one available to test sorption thermodynamics models. However, by FTIR analysis, we are able to provide also quantitative estimates for the amount of the different types of hydrogen bonding interactions established in the system. So we tested the model also against these data, since the theory, once its parameters have been determined by fitting of gravimetric isotherms, is able to provide predictions for these measurables ( see the new Appendix section added in the amended version of the manuscript). Hence, while figure 13 represents a data fitting, figure 14 reports the model predictions against the amounts estimated by spectroscopic means.

Reviwer #2: 

Lastly, the authors did not provide any mathematical equations which are of utmost importance for this manuscript. Although the reports have been cited in this manuscript, without any explicit indication, it is practically impossible to understand what the authors referred to.

Authors reply:

In the revised version of the manuscript an appendix section has been added where models equations  are thoroughly discussed.

Reviewer 3 Report

The manuscript entitled "Sorption thermodynamics of low molecular weight penetrants in a glassy polymer: a molecular perspective" describes the sorption thermodynamics of low molecular weight penetrants in glassy polymers endowed with specific interactions. Overall, the manuscript is well-written. This referee only has the following concerns.

1- The title is a little bit big since the low molecular weight penetrants cover more than the authors presented in this submission. Also, PEI is one polymer, cannot cover all glassy polymers.

2- The referee suggests enlarging the letters in Figure 1-3.

3- The unique findings are not clear. The authors should present which finding is different from previous reports.

Author Response

Reviewer #3:

The manuscript entitled "Sorption thermodynamics of low molecular weight penetrants in a glassy polymer: a molecular perspective" describes the sorption thermodynamics of low molecular weight penetrants in glassy polymers endowed with specific interactions. Overall, the manuscript is well-written. This referee only has the following concerns.

1- The title is a little bit big since the low molecular weight penetrants cover more than the authors presented in this submission. Also, PEI is one polymer, cannot cover all glassy polymers.

Authors reply:

The title has been modified according to the reviewer suggestion. It now reads: ‘Sorption thermodynamics of CO2, H2O and CH3OH in a glassy polyetherimide: a molecular perspective’.

Reviewer #3:

2- The referee suggests enlarging the letters in Figure 1-3.

Authors reply: 

Figures and letters have been enlarged.

 Reviewer #3:

3- The unique findings are not clear. The authors should present which finding is different from previous reports.

Authors reply:

As reported in the reply to reviewer #1, this point has been clarified by adding some sentences at the end of the introduction section detailing the new and previous findings. 

Reviewer 4 Report

The manuscript entitled: "Sorption thermodynamics of low molecular weight penetrants in a glassy polymer: a molecular perspective" is a very nice work, balancing the simulation together with experimental results for the sorption of low molecular weight substances in a glassy polymer.

I can recommend it for acceptance as it is since there are no scientific problems detected. It would be nice to expand these results into other polymers or even into cases that there is high exposure to organic solvents in order to link this resutsl with the performance of the polymers and their aging behavior.

Author Response

Reviewer #4:

The manuscript entitled: "Sorption thermodynamics of low molecular weight penetrants in a glassy polymer: a molecular perspective" is a very nice work, balancing the simulation together with experimental results for the sorption of low molecular weight substances in a glassy polymer.

I can recommend it for acceptance as it is since there are no scientific problems detected. It would be nice to expand these results into other polymers or even into cases that there is high exposure to organic solvents in order to link this results with the performance of the polymers and their aging behavior.

Authors reply: 

We thank the reviewer for the nice remark. Work is in progress along the lines suggested by the reviewer. The adopted techniques are well suited for the task. 

Round 2

Reviewer 1 Report

The authors have replied but slightly to some of the previous reviewing comments and reordered a little bit the manuscript. Still there are some concerns that have not been addressed as:

-novelty

“The present contribution puts together unpublished results on the CO2-PEI system with results on the H2O-PEI and CH3OH-PEI systems obtained by our group and already reported in recent publications [46-49].” (lines 61-63)

-ftir analyses

I suggest the authors make sure to deconvolute the saturation by thickness, water vapor and CO2 before jumping to conclusions. Figure 4 can hardly provide any information in the measurement conditions.

Please revise the Ënglish correctness because the expression is quite defective and contradictions observed at times that make the reading difficult.

Author Response

We have made extensive changes to the manuscript and added new content and figures, in a final attempt to make more clear the points that are apparently obscure to the reviewer #1 .

1) The research design and the followed methods are now described in greater details.

2) The results are now discussed more thoroughly and in a clearer manner. In particular the spectroscopic sections, that were perhaps too concise, have been extended and improved, also by adding 5 figures (new 5A, new 11A,B, new 16A,B).

3) Novelty of the contribution is explicitly stated and cannot be modified without completely altering the scope of our submission.  

 4) Regarding the comment of the reviewer:

“I suggest the authors make sure to deconvolute the saturation by thickness, water vapor and CO2 before jumping to conclusions. Figure 4 can hardly provide any information in the measurement conditions.”

 the sense of the first part of the comment is obscure to us. Regarding Figure 4, we now provided a better description of it as well as clarified its meaning.

5) The English has been improved to the best of our capabilities.

Reviewer 3 Report

The referee satified with the revision, and no further revision is needed.

Author Response

We tan the reviewer for the comment

Round 3

Reviewer 1 Report

The authors have reviewed the introduction, experimental and results discussion sections and the analysis of results is more clear than before.

I still do not see the point of Figure 4 where there are so many saturated peak bands in the spectra, that complicates the analysis of the bands that are remarked by the authors. That is why I insist that the authors try measurements at lower film thicknesses and provide replicability data in order to assure reproducibility.

Author Response

In the previous version of the manuscript, we have added some sentences in section 2.3 from which it was clear, at least for us, that spectroscopic analysis has been actually performed on both on 'thick' and 'thin' samples, stating also the motivation for that. 

We have now added a new figure (figure 4B) reporting the measured absorbance spectra of fully dried PEI (red trace) and PEI equilibrated at 35°C at a CO2 pressure of 150 Torr, for the case of a thin film. Moreover, in section 4.1.1, the issue of spectroscopic analysis of ‘thick’ and ‘thin’ films is further discussed, deepening the arguments already reported in section 2.3.